# The Impact of Backbone Fluorination and Side-Chain Position in Thiophene-Benzothiadiazole-Based Hole-Transport Materials on the Performance and Stability of Perovskite Solar Cells

**DOI:** 10.3390/ijms232113375

**Published:** 2022-11-02

**Authors:** Marina M. Tepliakova, Ilya E. Kuznetsov, Aleksandra N. Mikheeva, Maxim E. Sideltsev, Artyom V. Novikov, Aleksandra D. Furasova, Roman R. Kapaev, Alexey A. Piryazev, Artur T. Kapasharov, Tatiana A. Pugacheva, Sergei V. Makarov, Keith J. Stevenson, Alexander V. Akkuratov

**Affiliations:** 1Center for Energy Science and Technology (CEST), Skolkovo Institute of Science and Technology, Nobel St. 3, 143026 Moscow, Russia; 2Federal Research Center of Problems of Chemical Physics and Medicinal Chemistry, Russian Academy of Sciences, FRC PCPMC RAS, Academician Semenov Avenue 1, 142432 Chernogolovka, Russia; 3School of Physics and Engineering, ITMO University, Kronverksky Pr. 49, 197101 St. Petersburg, Russia; 4Department of Chemistry, Bar Ilan University, Ramat Gan 5290002, Israel; 5Bar-Ilan Institute of Nanotechnology and Advanced Materials, Ramat Gan 5290002, Israel; 6Department of Chemistry, Lomonosov Moscow State University, GSP-1, 1 Leninskiye Gory, 119991 Moscow, Russia; 7Sirius University of Science and Technology, Olympic Ave, 1, 354340 Sochi, Russia; 8Harbin Engineering University, Harbin 150001, China; 9Qingdao Innovation and Development Center of Harbin Engineering University, Qingdao 266000, China

**Keywords:** perovskite solar cells, stability, hole-transport materials, small molecules, benzothiadiazole

## Abstract

Perovskite solar cells (PSCs) currently reach high efficiencies, while their insufficient stability remains an obstacle to their technological commercialization. The introduction of hole-transport materials (HTMs) into the device structure is a key approach for enhancing the efficiency and stability of devices. However, currently, the influence of the HTM structure or properties on the characteristics and operational stability of PSCs remains insufficiently studied. Herein, we present four novel push-pull small molecules, **H1-4**, with alternating thiophene and benzothiadiazole or fluorine-loaded benzothiadiazole units, which contain branched and linear alkyl chains in the different positions of terminal thiophenes to evaluate the impact of HTM structure on PSC performance. It is demonstrated that minor changes in the structure of HTMs significantly influence their behavior in thin films. In particular, **H3** organizes into highly ordered lamellar structures in thin films, which proves to be crucial in boosting the efficiency and stability of PSCs. The presented results shed light on the crucial role of the HTM structure and the morphology of films in the performance of PSCs.

## 1. Introduction

Perovskite solar cells (PSCs) are a promising photovoltaic technology providing spectacular power conversion efficiencies (PCEs) of more than 25%, which already rivals the certified efficiencies of crystal-silicon photovoltaics [1]. The perovskite active layer can be fabricated with solution processing, thus paving the way for the assembly of lightweight and flexible photovoltaics using low-cost printing methods compatible with roll-to-roll technologies [2]. The bottleneck for the industrial-scale commercialization of PSCs remains the poor stability of the devices arising from various intrinsic and extrinsic factors [3]. The extrinsic factor is the sensitivity of the perovskite material to oxygen and the moisture of the atmosphere [4]. To avoid direct contact between the air and perovskite absorbers, proper encapsulating materials can be applied [5,6]. As for intrinsic degradation processes leading to PSC performance deterioration, they are more numerous and practically inevitable. It is well-studied that perovskite material decomposes under exposure to light and elevated temperatures with the formation of volatile products, such as I_2_, CH_3_NH_2_, etc. [7]. The migration of these decomposition products from the device structure leads to the degradation of the photo-active layer. The conservation of the volatile products inside the photo-active layer might improve the long-term stability of PSCs [8,9]. Therefore, the hole-transport layer, which is situated atop the perovskite in the standard n-i-p configuration, should possess low gas permeability [10,11,12]. In recent work, it was demonstrated that the application of a double hole-transport layer comprising a combination of organic material and transition metal oxides in the highest oxidation state, such as molybdenum (VI) oxide, vanadium (V) oxide, and tungsten (VI) oxide allows the achievement of outstanding stabilities of PSCs up to 4500 h under operation conditions [13]. Furthermore, in our work it was revealed that, while double hole-transport layers with vanadium oxide perform as a perfect encapsulating layer, it is the organic component, which should be optimized further to improve the performance of PSCs [14].

Conjugated organic small molecules are widely used as HTMs due to their well-defined structure and tunable electronic properties. One of the requirements for the HTM is the alignment of its highest occupied molecular orbital (HOMO) energy with the perovskite valence band, which ensures an energetically favorable hole-transport mechanism. On the other hand, the lowest unoccupied molecular orbital (LUMO) energy of the HTM should be significantly higher than the conduction band of the perovskite material in order to hinder electron transfer. 

A plethora of HTMs developed in recent years are based on either strong or weak electron donor blocks connected through π-conjugated bridges. Such moieties, including electron-rich arylamines, carbazole and fluorene derivatives, benzodithiophene, benzotrithiophene, phenothiazine, fused polycyclic cores, and triazine derivatives, are widely used for the design of small molecule HTMs [15,16,17,18,19,20,21].

However, for fine-tuning the electronic properties of conjugated molecules, the alternation of electron donor and electron acceptor fragments in the backbone are essential. Additionally, side chain engineering enables tailoring the morphology of HTM thin films, which is responsible for charge transport [22,23].

Herein, we present four novel conjugated small molecules with DADAD structures (a D-donor thiophene block and an A–acceptor benzothiadiazole block) and their investigation as HTMs in perovskite solar cells (Figure 1). 

The presented materials incorporate thiophene rings bearing 2-ethylhexyl and n-octyl substituents in the α and β positions for **H1-2**, and in the β and α positions for **H3-4**, respectively. The variation of the side chain positions on the thiophene rings and the introduction of fluorine to the benzothiadiazole moieties allowed us to study the relationships between the structure of DADAD-type HTMs and its film-forming properties affecting the performance and operational stability of PSCs based on them.

## 2. Results and Discussion

The syntheses of the materials **H1-4** are presented in Figure 2. In brief, compounds **2a-b** were prepared via the palladium-catalyzed Stille cross-coupling reaction between 4,7-dibromobenzo[c]-1,2,5-thiadiazole or 5,6-difluoro-4,7-dibromobenzo[c]-1,2,5-thiadiazole and stannanes **1a-b**. Mono-functionalized products **1a-d** were isolated, purified, and used for the synthesis of the target compounds via the Stille reaction using 2,5-bis(tributylstannyl)thiophene. All compounds exhibited good solubility in common organic solvents, such as dichloromethane, chloroform, and toluene, which enables their deposition in multi-layered organic electronic devices from solutions.

The detailed synthetic and purification procedures are described in more detail in the materials and methods section.

In the next step, the physico-chemical properties of the obtained materials were investigated. The thermal stability of the synthesized molecules was studied using thermal gravimetric analysis (TGA) performed under an inert atmosphere. The decomposition temperatures corresponding to a 5% weight loss ranged from 357 °C to 395 °C, indicating that all compounds possess sufficient thermal stability for their application in PSCs (Appendix A). The differential scanning calorimetry (DSC) measurements revealed melting temperatures of 152 °C and 154 °C and crystallization temperatures of 140 °C and 142 °C for fluorine-loaded **H2** and **H4**, respectively. The broad peaks at 53 °C and 51 °C can be observed on the DCS plots for **H1** and **H3**, respectively, implying their more amorphous nature compared to **H2** and **H4**.

In order to obtain more insight into the crystallinity of materials **H1**-**4** in thin films, X-ray diffraction patterns (XRD) were collected (Figure 3a and Appendix A). The XRD patterns of all materials exhibited sharp peaks in the small-angle region with d-spacings of >25Å, which indicates that the ordering of the molecules was in the liquid crystal phase [24]. The intense reflexes for the **H2** and **H4** samples at 2θ = 9.6–11.6° imply the superior crystallinity of films based on fluorine-containing compounds. This type of order may be attributed to the tilted organization of molecules in thin films [24,25]. The broad wide-angle peaks at 2θ = 24–30° are typical for the π-π stacking of conjugated molecules.

The optoelectronic and charge-transport properties of the materials were characterized using absorption spectroscopy, cyclic voltammetry, and the space-charge-limited current measurements technique, as described before [26]. In summary, all materials provided suitable HOMO energy levels lying in the range of −5.4 ± 0.2 eV and decent hole mobilities of 10^−3^–10^−4^ cm^2^ V^−1^ s^−1^ (Appendix A), which are acceptable for their applications as HTMs for PSCs. The bandgaps (E_g_) estimated from the low-energy absorption bands in the optical spectra of the thin films using the Tauc method are in the range of 1.9–2.0 eV (Appendix A) [27]. This little difference in E_g_ values is defined primarily by the similar length of the conjugated backbone of the obtained materials.

In the next step, materials **H1-4** were investigated in PSCs with n-i-p configurations reported previously [28]. The detailed procedures of the devices’ fabrications are provided in the materials and methods section. Briefly, the glass with the conductive layer of indium-doped tin oxide (ITO) was used as an electron-collecting electrode and was covered with an ETM SnO_2_, which was further passivated with the acidic fullerene derivative, phenyl-C_61_-butyric acid [29]. The photo-active perovskite layer, MAPbI_3,_ was deposited atop an inert atmosphere. Furthermore, the **H1-4** HTMs were spin-coated from the solutions in chlorobenzene. Finally, the MoO_x_ and silver electrodes were thermally evaporated under a vacuum, leading to the final ITO/SnO_2_:PCBA/MAPbI_3_/**H1-4**/MoO_x_/Ag architecture shown in Figure 4a. 

The current-voltage characteristics of the PSCs with **H1-4** as HTMs are summarized in Table 1, and the best forward and reverse scans are providedin Figure 4b. The current density values were confirmed with external quantum efficiency spectra (Figure 4c).

Non-fluorinated HTMs **H1** and **H3** delivered higher efficiencies in the devices with power conversion efficiencies of 12.0% and 12.6%, respectively, outperforming the devices with **H2** and **H4** in voltages (V_OC_), fill factors (FF), and current densities (*J_SC_*). Low molecular weight organic HTMs are usually applied with dopants to enhance their charge-transport characteristics at the price of device stability mitigation [30,31]. It is worth noting that the presented results were achieved for undoped HTMs. The reference cells with non-doped spiro-OMeTAD as an HTM were also fabricated. They reached a lower efficiency of 7.0 ± 2.0% with V_OC_ = 910 ± 60 mV, *J*_SC_ = 16.1 ± 0.5 mA cm^−2^, and FF = 46 ± 10% (Appendix A). These results are in agreement with previously reported findings, which highlight the potential of the designed oligomers (**H1-4**) as HTMs [32,33].

To obtain more insight into the possible reasons for the differences in the characteristics of the devices, the steady-state photoluminescence (PL) spectra of samples with configuration glass/MAPbI_3_/HTM were collected (Figure 4d) [34]. The more pronounced PL quenching of MAPbI_3_/**H1** and MAPbI_3_/**H3** was observed, which suggests better hole extraction from perovskite to non-fluorinated HTMs.

Furthermore, the surface of the tri-layer stacks with the configuration glass/perovskite/**H1**-**4**/metal oxide was investigated using atomic force microscopy (AFM) (Figure 5a). The introduction of the metal oxide, in this case, allowed us to study the behavior of the double organic-inorganic HTM, as it is presented in the structure of the photovoltaic devices. Previously, it was demonstrated that the layer formed by the metal oxide is homogeneous and smooth; thus, the morphology of the whole tri-layer stack depends on the nature of the underlying layer [13]. 

As can be seen from the AFM scans, the samples with **H3** represent an organized texture with lamellar features, which resembles the morphology of films based on the poly(3-hexylthiophene) (P3HT) polymer, a state-of-the-art p-type semiconductor [35]. 

In the case of P3HT, its organization in thin films into ordered domains positively influences the charge-transfer characteristics, which is similar to what we have with **H3** and explains the superior characteristics of PSCs based on **H3**. The surface of **H1** contains large formations and cavities all along the surface. In the AFM images of the samples with **H2** and **H4,** formations of a 0.5–0.7 µm size can be seen, which are close in size to the grains of perovskite (Appendix A), with the voids between them. Such voids may be the reason for the lower voltages due to the reduced contact of the adjacent layers. Additionally, the defects may increase the rate of trap-assistant recombination [36]. Thus, the film-forming properties and morphology of HTMs are important parameters which impact the performance of PSCs. Particularly, only material **H3** consisting of a combination of non-fluorinated benzothiadiazole and terminal 2-octyl-3-(2-ethylhexyl)thiophenes enables well-organized self-assembling in thin films, and thus the superior efficiency and stability of perovskite solar cells. As mentioned before, according to the DSC, **H1** and **H3** possess a low-temperature phase transition at 51–53 °C, which lies in the operating temperature range of solar cells at 50–60 °C [37]. Therefore, it was interesting to track the changes occurring in the samples after exposure to elevated temperatures. The morphology of the tri-layer stacks with the configuration of glass/perovskite/**H1**-**4**/metal oxide after 10 min annealing at 60 °C was characterized (Figure 5a). 

The annealed film **H4** exhibited insignificant changes in the surface morphology. Notably, the number of surface cavities on the **H2**-based substrate substantially decreased, which indicates self-healing processes occurring during heating. The lamellar structure of the sample with **H3** remained unaffected. The most remarkable changes occurred in the morphology of the sample with **H1**, which turned into a net-like structure.

The degree of order might have a positive impact on the efficiency of the devices due to enhanced charge-transfer properties. At the same time, penetrating surface defects may act as pathways for the volatile decomposition products. Therefore, it was very important to study the influence of HTM morphology on the stability of solar cells. 

In the next step, the stability of PSCs incorporating **H1**-**4** as HTMs was evaluated in an inert atmosphere under the constant illumination of 75 ± 3 mW/cm^2^ and a temperature of 45 ± 3 °C. It is well-known that MAPbI_3_ is unstable under light and temperature because of the high reactivity of its component CH_3_NH_3_I (MAI) [38,39]. Therefore, for this experiment, devices with another configuration were fabricated. They consisted of more stable Cs(NH_2_)_2_CHPbI_3_ perovskite, ZnO as ETL, an Al electrode, and vanadium oxide as an inorganic component of the double hole-transport layer [14]. The evolution of the device efficiency under constant illumination is presented in Figure 5b. 

The efficiency of the devices with **H2** and **H3** improved since the starting point and remained unchanged during the experiment, while for devices with **H1** and **H4,** the trend in the slow efficiency decrease was detected. This phenomenon may be related to the substantial amount of deep surface defects discovered for the **H1** and **H4** tri-layer stacks using AFM. 

In summary, the coupling of the non-fluorinated benzothiadiazole block (B) with thiophene (T) and alkylthiophene blocks is the preferable synthetic route for the design of TBTBT-family small molecule HTMs for perovskite solar cells, as such combination promotes the organization of highly ordered films. A further boost of the PSC efficiency might be reached by designing HTMs with **H3** as a basis in combination with other prospective heteroaromatic blocks.

## 3. Materials and Methods

### 3.1. Materials and Instruments

All solvents and reagents were purchased from Sigma-Aldrich or Acros Organics and were used as received or purified according to standard procedures. 

Mass spectra were obtained using the electrospray/APCI combined ionization technique and a GC-MS LCMS-2020 instrument.

The optical properties of the compounds were investigated using a SPECS SSP-705-1 scanning spectrophotometer. 

The NMR ^1^H,^13^C, and ^19^F spectra were recorded on a Bruker AVANCE 500 instrument. 

The thermal gravimetry analysis (TGA) and differential scanning calorimetry (DSC) for the compounds were obtained using Q50 TA and Netzsch DSC 214 Polyma instruments, respectively. All measurements were carried out under argon with a heating rate of 10 °C min^−1^. 

The electrochemical properties of oligomers **H1**-**4** were studied with cyclic voltammetry (CV). CV measurements were carried out for the thin films of the oligomers deposited from the chloroform solutions on a glassy carbon electrode (CH Instruments Inc. (Austin TX, USA), d = 5 mm). The solution of Bu_4_NBF_4_ in acetonitrile (0.1 M) was used as a supporting electrolyte, a platinum wire was used as a counter electrode, and Ag/AgCl (in 4M KCl) was used as a reference electrode (CH Instruments Inc.). A ferrocene/ferrocenium couple was used as an internal standard. The CV curves were recorded at room temperature using an ELINS P-20-X instrument (the potential sweep rate was 50 mV s^−1^).

The external photon-to-current conversion efficiency spectra were performed via the commercial apparatus Asani Spectra MAX-303, with a monochromator (Asani Spectra CMS-100) coupled with a xenon lamp. A mono-crystalline silicon solar cell (15150-KG5, ABET technologies, Milford, CT, USA) connected to a Keithley 2400 calibrator-multimeter was used as a reference cell.

An X-ray diffraction analysis was performed using an X-ray Powder Diffractometer Huber G670 with a CoKα radiation linear PSD detector.

The morphology of the tri-layer stacks was characterized using an NTEGRA PRIMA (NT-MDT, Moscow, Russia) atomic force microscope.

The hole mobilities were estimated using the SCLC technique. The hole-only diodes were fabricated on ITO-glass substrates. Then the PEDOT:PSS (PH 1000, Heraeus Clevios, Hanau, Germany) was deposited by spin-coating followed by drying the substrates at 150 °C for 15 min. The films of **H1**-**H4** were formed by spin-coating toluene solutions (10 mg/mL). After that, MoO_3_ and silver electrodes were successively deposited in a vacuum chamber (6 × 10^−6^ mbar), resulting in devices with the architecture as follows: ITO/PEDOT:PSS/**H1**-**H4**/MoO_3_/Ag. The film thickness was recorded using AFM spectroscopy when scanning a thin scratch on the top of the films.

The *J*-V curves were recorded in the glovebox under the illumination (100 mW/cm^2^) provided by the Newport Verasol AAA solar simulator using Advantest 6240A source-measurement units.

### 3.2. Fabrication of the Solar Cells with ***H1-4*** for the Efficiency Investigation

The bottom electrodes, composed of glass covered with conductive indium oxide doped with tin (ITO, 15 Ω/sq., Kintec, Kowloon, Hong Kong), were preliminarily treated by ultrasonication in acetone, water, isopropanol, and air-plasma (50% power, 5 min). A total of 10% of the suspension of SnO_2_ nanoparticles in water (Alfa Aesar) was spin-coated at 4000 rpm in two steps, and each layer was left drying for 20 s. Further, samples were placed in a cold heater heated to 175 °C for 10 min and annealed at this temperature for 30 min. The passivation layer for SnO_2_ (PCBA, 0.2 mg/mL in chlorobenzene (CB)) was spin-coated statically in the glove box. The samples were annealed at 100 °C for 10 min. The solution of MAPbI_3_ perovskite (1.4 M) was prepared by dissolving equimolar amounts of CH_3_NH_3_I and PbI_2_ in a mixture of dimethylformamide (80%) and n-methylpyrrolidone (20%) to give the 1.4M solution of perovskite ink. The perovskite was spin-coated at 3000 rpm dynamically and left to dry for 20 min. In the next step, samples were heated to 80 °C for 10 min and annealed at this temperature for 5 min. Solutions of **H1-4** (6 mg/mL in CB) were spin-coated at 3000 rpm and were preliminarily confirmed as the optimal deposition conditions for each material. The layer of MoO_x_ (10 nm) was thermally evaporated on the entire area (under 10^−5^ mbar). Finally, the silver top electrode (100 nm) was deposited using thermal evaporation (under 10^−5^ mbar) through the shadow mask, defining the area of the final cells as 0.1 cm^2^. 

### 3.3. Fabrication of the Solar Cells with ***H1-4*** for the Stability Evaluation

The conductive ITO substrates were pre-cleaned with ultrasonication and air plasma, as described before. The ETL solution was prepared by dissolving Zn acetate (100 mg) in the mixture of monoethanolamine and 2-methoxyethanol (33 μL and 1 mL, respectively) and deposited on the ITO using spin-coating. Then, the samples were slowly heated to 200 °C and annealed at this temperature for 1 h. The MAI solution (8 mg/mL in isopropanol) was spin-coated on ZnO, annealed at 200 °C for 5 min, heated to 300 °C, and cooled to room temperature. The PCBA passivation (0.2 mg/mL in CB) was deposited atop ZnO using spin-coating. The Cs(NH_2_)_2_CHPbI_3_ ink with a 1.4M concentration was prepared from CsI, (NH_2_)_2_CHI, and PbI_2_ in DMF and DMSO 85% and 15%, respectively, and then spin-coated at 4000 rpm and instantly annealed at 100 °C for 10 min. The solutions of **H1-4** in chlorobenzene (6 mg/mL) were spin-coated atop the perovskite film. The layer of VO_x_ (30 nm) was thermally evaporated on the entire area (under 10^−5^ mbar). Finally, an aluminum top electrode (100 nm) was deposited using thermal evaporation through the shadow mask, defining the area of the final cells as 0.1 cm^2^. 

### 3.4. Fabrication of the Tri-layer Stacks for the AFM Measurements

The glass substrates were cleaned by ultrasonication in acetone, water, and isopropanol and treated with air plasma for 5 min. The MAPbI_3_ perovskite 1.4M in DMF:NMP was spin-coated on the glass substrate. The layers of **H1**-**4** were spin-coated atop the chlorobenzene solutions (6 mg/mL). The stacks were covered with VO_x_ and thermally evaporated under reduced pressure. Half of the samples were annealed at 60 °C.

### 3.5. Synthesis of Compound ***2a***

Compound **1a** (16.04 g, 34 mmol) and 4,7-dibromobenzo[c][1,2,5]thiadiazole (10.0 g, 34 mmol) were dissolved in toluene (60 mL) under argon in a two-necked flask. Then, tetrakis(triphenylphosphine)palladium(0) (8 mg) was added. The mixture was refluxed within 24 h and then cooled to room temperature. The solvent was removed on a rotary evaporator, and the obtained crude product was purified by column chromatography on silica using a hexane:toluene mixture as the eluent. The yield of **2a** was 35–37%. ^1^H NMR (CDCl_3_, 500 MHz, δ): 7.86 (s, 1H), 7.77 (d, 1H), 7.60(d, 1H), 2.68 (d, 2H), 2.53 (t, 2H), 1.60 (m, 3H), 1.40–1.27 (m, 18H), 0.86 (m, 9H), ppm. 

### 3.6. Synthesis of Compound ***2b***

Compound **2b** was synthesized according to the procedure described for **2a** using 5,6-difluoro-4,7-dibromobenzo[c][1,2,5]thiadiazole (10 g, 30.4 mmol) and compound **1a** (14.28 g, 30.4 mmol). The yield of **2b** was 36–38%. ^1^H NMR (CDCl_3_, 500 MHz, δ): 8.00 (s, 1H), 2.72 (d, 2H), 2.56 (t, 2H), 1.60 (m, 2H), 1.35–1.24 (m, 18H), 0.85–0.90 (m, 9H), ppm; ^19^F NMR (CDCl_3_, 470 MHz, δ): −120.47, −127.87 ppm.

### 3.7. Synthesis of Compound ***2c***

Compound **2c** was synthesized according to the procedure described for **1a** using 4,7-dibromobenzo[c][1,2,5]thiadiazole (10.0 g, 34 mmol) and compound **1b** (16.04 g, 34 mmol). The yield of **2c** was approximately 39–41%. ^1^H NMR (CDCl_3_, 500 MHz, δ): 7.80 (s, 1H); 7.77 (d, 2H); 7.59 (d, 2H); 2.74–2.77 (t, 2H), 2.48 (m, 2H), 1.69 (m, 2H), 1.40–1.24 (m, 18H), 0.85–0.90 (m, 9H), ppm. 

### 3.8. Synthesis of Compound ***2d***

Compound **2d** was obtained according to the synthetic procedure described for **1a** using 5,6-difluoro-4,7-dibromobenzo[c][1,2,5]thiadiazole (10 g, 30.4 mmol) and compound **2b** (14.28 g, 30.4 mmol). The yield of **2d** was 38–40%. ^1^H NMR (CDCl_3_, 500 MHz, δ): 7.93 (s, 1H), 2.78 (t, 2H), 2.51 (d, 2H), 1.70–1.72 (m, 2H), 1.40–1.26 (m, 18H), 0.84–0.90 (m, 9H), ppm. ^19^F NMR (CDCl_3_, 470 MHz, δ): −120.48; −128.04.

### 3.9. Synthesis of ***H1***

Compound **H1** was synthesized by the Stille cross-coupling reaction of **2a** (10.44 g, 20 mmol) with 2,5-bis(tributylstannyl)thiophene (6.62 g, 10 mmol). All reagents were dissolved in toluene (60 mL) in a three-necked round-bottom flask. The flask was filled with argon, and Pd(PPh_3_)_4_ (0.011 g, 0.01 mmol) was added. The mixture was refluxed for 24h, cooled to room temperature, and excess ethanol was added. The precipitate was collected by filtration, and the crude product was purified by column chromatography on silica using hexane:toluene as the eluent. The yield of pure **H1** was ca. 60%. ^1^H NMR (CDCl_3_, 500 MHz, δ): 8.18 (s, 2H), 7.93 (m, 4H), 7.78 (d, 2H), 2.72 (d, 4H), 2.56 (t, 4H), 1.65–1.67 (m, 6H), 1.30–1.37 (m, 36H), 0.93 (m, 18H) ppm; ^13^C NMR (CDCl_3_, 126 MHz, δ): 152.65, 152.56, 140.63, 140.40, 140.08, 134.96, 129.75, 128.10, 126.50, 125.73, 124.81, 124.71, 41.78, 32.61, 32.32, 31.89, 30.88, 29.62, 29.50, 29.29, 28.89, 28.51, 25.70, 23.04, 22.66, 14.13, 14.09, 10.88 ppm. ESI, *m*/*z* = 963.5 ([M-H]^−^).

### 3.10. Synthesis of ***H2***

The synthesis of compound **H2** was performed according to the procedure described for **H1**. The Stille coupling reaction of **2b** (11.06 g, 20 mmol) with 2,5-bis(tributylstannyl)thiophene (6.62 g, 20 mmol) afforded **H2** as red crystals. The yield was 58%.^1^H NMR (CDCl_3_, 500 MHz, δ): 8.17 (s, 2H), 7.94 (s, 2H), 2.65 (d, 4H), 2.49 (t, 4H), 1.59 (m, 6H), 1.39–1.28 (m, 36H), 0.90 (m, 18H) ppm; ^13^C NMR (CDCl_3_, 126 MHz, δ): 148.76, 148.69, 143.12, 139.46, 134.71, 132.95, 130.40, 127.48, 112.32, 112.23, 110.14, 110.04, 41.70, 32.63, 32.19, 31.95, 30.89, 29.71, 29.55, 29.37, 28.89, 28.34, 25.73, 23.09, 22.73, 14.18, 14.15, 10.87 ppm; ^19^F NMR (CDCl_3_, 470 MHz, δ): −126.27, −129.01 ppm. ESI, *m*/*z* = 1035.4 ([M-H]^−^).

### 3.11. Synthesis of ***H3***

Compound **H3** was synthesized by the Stille coupling reaction between compound **2c** (7.72 g, 14 mmol) and 2,5-bis(tributylstannyl)thiophene (4.64 g, 7 mmol) according to the procedure described for **H1**. The product **H3** was obtained as a dark red powder with a yield of 55%. ^1^H NMR (CDCl_3_, 500 MHz, δ): 8.17 (s, 2H), 7.91 (d, 2H), 7.84 (s, 2H), 7.77 (d, 2H), 2.76 (t, 4H), 2.50 (d, 4H), 1.71 (m, 4H), 1.40–1.24(m, 36H), 0.90–0.86 (m, 18H) ppm; ^13^C NMR (CDCl_3_, 126 MHz, δ): 152.67, 152.61, 142.38, 140.42, 138.32, 134.47, 130.32, 128.13, 126.59, 125.77, 124.83, 124.72, 40.62, 32.66, 32.60, 31.87, 31.85, 29.48, 29.40, 29.24, 28.89, 28.24, 25.74, 23.12, 22671, 14.14, 14.10, 10.92 ppm. ESI, *m*/*z* = 963.5 ([M-H]^−^).

### 3.12. Synthesis of ***H4***

Compound **H4** was synthesized by the Stille coupling reaction between compound **2d** (9.28 g, 16 mmol) and 2,5-bis(tributylstannyl)thiophene (5.30 g, 8 mmol) according to the procedure described for **H1**. The pure **H4** was obtained as red crystals with a yield of 59%. ^1^H NMR (CDCl_3_, 500 MHz, δ): 8.24 (s, 2H), 7.91 (s, 2H), 2.72 (t, 4H), 2.46 (d, 4H), 1.69 (m, 4H), 1.36–1.29 (m, 36H), 0.91–0,86 (m, 18H) ppm; ^13^C NMR (CDCl_3_, 126 MHz, δ): 148.91, 148.83, 144.86, 137.81, 134.81, 133.83, 130.69, 126.91, 112.60, 112.51, 110.32, 110.22, 40.60, 32.58, 32.46, 31.87, 31.75, 29.54, 29.38, 29.24, 28.88, 28.15, 25.73, 23.104, 22.67, 14.15, 14.10, 10.93 ppm; ^19^F NMR (CDCl_3_, 470 MHz, δ): −126.40, −129.23 ppm.ESI, *m*/*z* = 1035.4 ([M-H]^−^).

## 4. Conclusions 

Four novel donor-acceptor pentamers, **H1**-**H4,** based on alternating benzothiadiazole moieties with and without fluorine loading and thiophenes with alkyl side chains in different positions were synthesized and applied as hole-transport materials in perovskite solar cells. It was shown that the only material (**H3**) consisting of a combination of non-fluorinated benzothiadiazole and terminal 2-octyl-3-(2-ethylhexyl)thiophenes enables the ordering of the material in thin films, and thus superior efficiency and stability in perovskite solar cells. Other possible combinations either form visible pinholes in hole-transport layers atop the perovskite, as was observed for **H1** and **H4**, or promote recombination on the interface with the perovskite in the case of **H2**. The material **H3** may be potentially used as a perspective building block for novel hole-transport materials for a further boost of PSC efficiency. The results of this work provide a good insight into the design of promising organic HTMs and feature the importance of morphological control towards the development of efficient and stable PSCs.

## Figures and Tables

**Figure 1 ijms-23-13375-f001:**
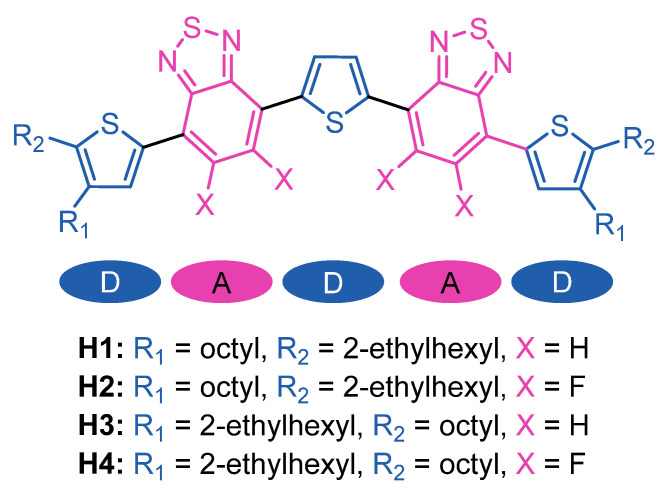
Molecular structures of designed **H1-4** molecules.

**Figure 2 ijms-23-13375-f002:**
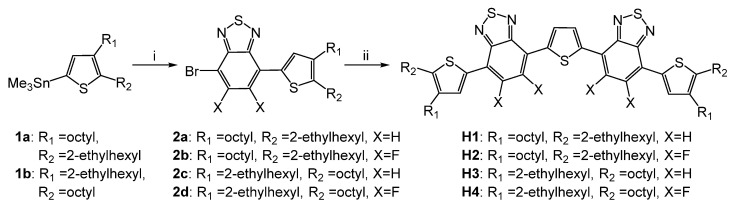
Synthesis of compounds **H1**-**4**. Conditions: i—4,7-dibromobenzo[c]-1,2,5-thiadiazole or 5,6-difluoro-4,7-dibromobenzo[c]-1,2,5-thiadiazole, Pd(PPh_3_)_4_, toluene, reflux; ii—2,5-bis(tributylstannyl)thiophene, Pd(PPh_3_)_4_, toluene, reflux.

**Figure 3 ijms-23-13375-f003:**
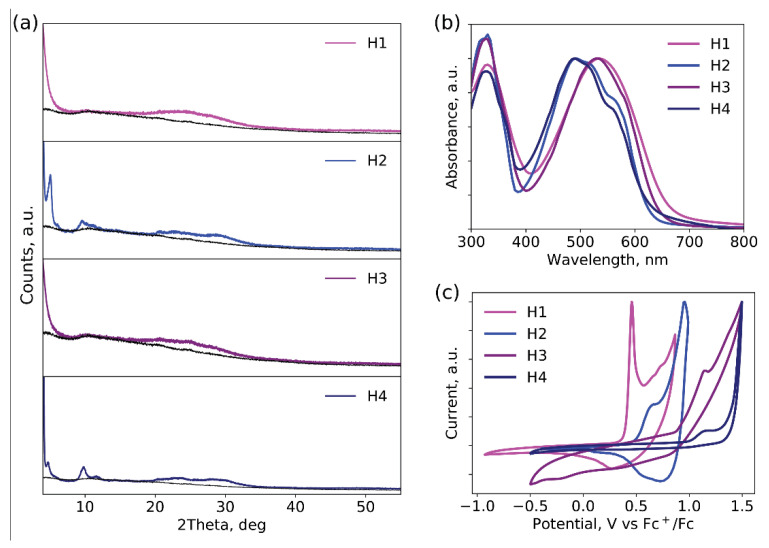
XRD patterns (**a**) absorption spectra (**b**) and cyclic voltammograms (**c**) of **H1**-**4** thin films.

**Figure 4 ijms-23-13375-f004:**
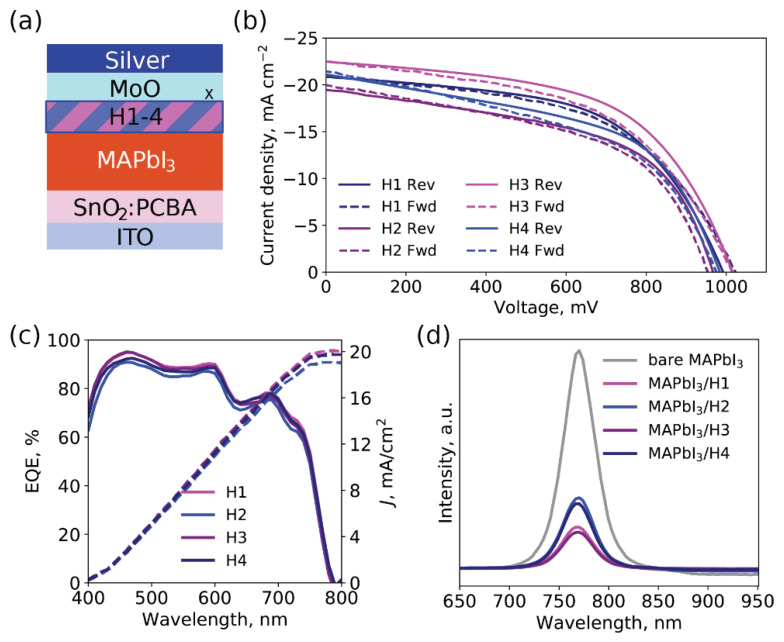
Configuration of PSCs (**a**), J-V curves of PSCs with **H1-4** (**b**), EQE spectra (**c**), and PL spectra for substrates with configuration glass/perovskite/HTM (**d**).

**Figure 5 ijms-23-13375-f005:**
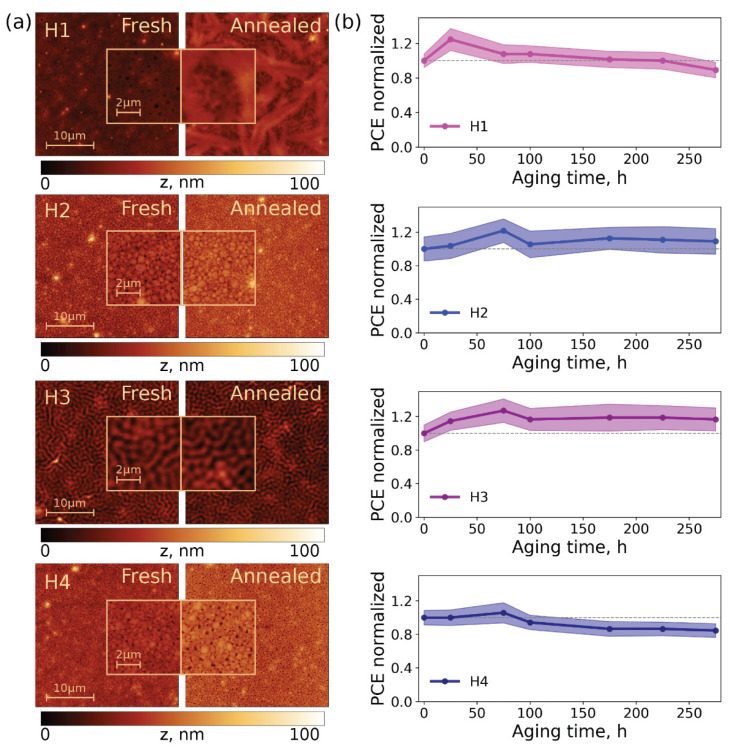
30 × 30 µm AFM scans (insets: the magnified 6 × 6 µm area) of freshly prepared and annealed at 60 °C tri-layer stacks glass/perovskite/**H1-4**/metal oxide (**a**); evolution of efficiency of PSCs with **H1-4** (**b**).

**Table 1 ijms-23-13375-t001:** Photovoltaic performance of PSCs based on **H1**-**4**.

HTM	V_OC_, mV	*J*_SC,_ mA cm^−2^	FF, %	PCE, %
**H1**	1020 * (970 ± 40)	22.3 (20.6 ± 0.7)	56 (48 ± 4)	12.0 (9.8 ± 1.2)
**H2**	960 (940 ± 10)	20.3 (19.7 ± 0.5)	53 (49 ± 1)	10.0 (9.3 ± 0.3)
**H3**	1020 (1020 ± 10)	22.9 (20.9 ± 0.8)	55 (50 ± 3)	12.6 (10.7 ± 0.9)
**H4**	1000 (980 ± 10)	21.6 (20.9 ± 0.8)	52 (46 ± 2)	10.8 (9.4 ± 0.8)

* maximal value (average ± standard deviation).

## Data Availability

The data presented in this study are available in this article.

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
