# Peer review of "The Impact of Backbone Fluorination and Side-Chain Position in Thiophene-Benzothiadiazole-Based Hole-Transport Materials on the Performance and Stability of Perovskite Solar Cells"

_ijms, 2022, doi:10.3390/ijms232113375_

Round 1
Reviewer 1 Report
Tepliakova et al. reported thiophene-benzothiadiazole-based HTMs for n-i-p planar perovskite solar cells. They synthesized four molecules with different side chains and terminations (replacing hydrogen with fluorine) and studied their influence on photovoltaic performance. The maximum PCE of 12.6% was achieved by using H3 molecule. The manuscript is well written and well organized, but it still can be improved from scientific aspect. The authors should consider below points and revise the manuscript accordingly.
1. The properties of molecules were characterized by DSC, TGA, CV, mobility etc. and their performance in device as HTL was examined. However, there is no explanation or discussion on why a particular molecule (in this case H3) gives the best device performance. Authors should include the structure-function relationship and relevant discission.
2. Fill factor (FF) of devices is quite low (even highest one is 56% only) compared to the reported perovskite solar cells. Please discuss why FF is lower than the state-of-the-art devices; could it be due to the HTL or other reasons?
3. Figure 3a and S2 are the same and one should be removed.
4. Please provide the J-V curve for the SCLC mobility test and measurement details.
5. The bandgap of the materials can be calculated from the absorption band edge. Authors should compare with the bandgap defined by length of conjugated backbone.
6. In addition to PL spectra, the time-resolved PL should be tested and calculate the decay time to explain the charge extraction at perovskite/HTL interface.
7. In the line 302, authors deduce the highly-ordered lamellar structure based on AFM image. AFM can only tell the morphology of the film and cannot reveal the crystal property such as lamellar structure. GWIAXS measurement is a proper tool to analyse the crystal structure.
8. There are many papers reported high-performance HTMs using various core units with triphenylamine arm units. Authors should include these high-performance HTMs in the introduction and cite. For example:
Solar RRL, vol. 3, 1900011 (2019)
J. Mater. Chem. A., vol. 7, 9510 (2019)
Solar RRL, vol. 4, 2000327 (2020)
Synthetic Metals, vol. 291, 117174 (2022)
Author Response
Tepliakova et al. reported thiophene-benzothiadiazole-based HTMs for n-i-p planar perovskite solar cells. They synthesized four molecules with different side chains and terminations (replacing hydrogen with fluorine) and studied their influence on photovoltaic performance. The maximum PCE of 12.6% was achieved by using H3 molecule. The manuscript is well written and well organized, but it still can be improved from scientific aspect. The authors should consider below points and revise the manuscript accordingly.
Thank you for your positive report and your suggestions!
- The properties of molecules were characterized by DSC, TGA, CV, mobility etc. and their performance in device as HTL was examined. However, there is no explanation or discussion on why a particular molecule (in this case H3) gives the best device performance. Authors should include the structure-function relationship and relevant discussion.
Thank you for this valuable suggestion! In the revised version of manuscript, we emphasized that the differences in optoelectronic properties of presented materials have a little impact on the performances of perovskite solar cells based on them. Thus, no strong correlations between HOMO energies, bandgaps or hole mobilities with the characteristics in photovoltaic devices were revealed. However, we showed that the structure features of oligomers influence their morphology in films (lines 338-342). This in turn, affects the photovoltaic characteristics of devices. Particularly, only material H3 consisting of combination of non-fluorinated benzothiadiazole and terminal 2-octyl-3-(2-ethylhexyl)thiophenes enables well-organized self-assembling in thin films, and thus superior efficiency and stability of perovskite solar cells.
- Fill factor (FF) of devices is quite low (even highest one is 56% only) compared to the reported perovskite solar cells. Please discuss why FF is lower than the state-of-the-art devices; could it be due to the HTL or other reasons?
Thank you for this comment! Generally, low FF are usually related to insufficient charge transfer characteristics, which is typical for undoped organic HTMs. [Energy Environ. Sci., 2021,14, 5161-5190; Adv.Energy Mater., 2018, 8(9), 1702512]. The hole mobilities of reported HTMs vary in the range of 10-4-10-3 cm2V-1s-1according to SCLC measurements. Another possible reason may lie in the defective interface between HTM and adjacent layers (perovskite and electrode). This defects on the interface may serve as charge traps, thus mitigating overall performance of the devices [ACS Energy Lett. 2021, 6, 11, 3916–3923]. Further improving of FFs can be addressed by passivating the perovskite defects at the interface between the perovskite and HTMs. Besides, the optimal material (H3) may serve as a building block for polymeric HTMs, which usually enables additional charge transfer pathways compensating low hole mobility.
- Figure 3a and S2 are the same and one should be removed.
Thank you! We removed the first part of Figure S2 and left only selected (magnified) regions of XRD patterns.
- Please provide the J-V curve for the SCLC mobility test and measurement details.
Thank you for this suggestion! We added J-V curve for hole-only devices in Supporting Information as Figure S3
- The bandgap of the materials can be calculated from the absorption band edge. Authors should compare with the bandgap defined by length of conjugated backbone.
The bandgaps of all compounds were estimated from low-energy absorption bands in optical spectra of thin films using Tauc method [J. Phys. Chem. Lett. 9 (23), 6814–6817]. The Eg values were estimated to be in the range of 1.9-2.0 eV. This insignificant difference in Eg is defined mostly by the similar length of conjugated backbone of obtained materials. We added corresponding comments in the main text regarding this issue to make this clear for readers (lines 276-278).
- In addition to PL spectra, the time-resolved PL should be tested and calculate the decay time to explain the charge extraction at perovskite/HTL interface.
We agree with Reviewer that time-resolved PL spectroscopy could provide more information on charge extraction at interface between perovskite material and HTM used. Unfortunately, we are not able to measure TRPL for our samples due to the lack of precise equipment. We believe this omission should have a little influence on the presented data
- In the line 302, authors deduce the highly-ordered lamellar structure based on AFM image. AFM can only tell the morphology of the film and cannot reveal the crystal property such as lamellar structure. GWIAXS measurement is a proper tool to analyse the crystal structure.
Thank you for this comment! We agree that AFM technique cannot evaluate crystallinity of thin films. This sentence was used to characterize the texture of organized films in case of H3. We revised this sentence in more accurate form: “As can be seen from AFM, the samples with H3 represent an organized texture with lamellar features, which resembles the morphology of films based on P3HT polymer, a state-of-the-art p-type semiconductor” (lines 323-325)
- There are many papers reported high-performance HTMs using various core units with triphenylamine arm units. Authors should include these high-performance HTMs in the introduction and cite. For example:
Solar RRL, vol. 3, 1900011 (2019)
- Mater. Chem. A., vol. 7, 9510 (2019)
Solar RRL, vol. 4, 2000327 (2020)
Synthetic Metals, vol. 291, 117174 (2022)
Thank you! We cited these papers in the Introduction.
Reviewer 2 Report
The authors studied thiophene and benzothiadiazole molecules as hole transport materials for perovskite solar cells. By controlling the structure of the molecules, the authors demonstrated high efficiency and stability in the PSCs for non-fluorinated benzothiadiazole and terminal thiophene molecules based HTM. The manuscript can be accepted for publication in its current form.
Author Response
Reviewer 2
The authors studied thiophene and benzothiadiazole molecules as hole transport materials for perovskite solar cells. By controlling the structure of the molecules, the authors demonstrated high efficiency and stability in the PSCs for non-fluorinated benzothiadiazole and terminal thiophene molecules based HTM. The manuscript can be accepted for publication in its current form.
Thank you for your positive evaluation!
Reviewer 3 Report
The design and synthesis of structurally similar derivatives (H1-H4) based on alternating thiophene and benzothiadiazole (with/without fluorine) with varying alkyl chain lengths at the terminal thiophenes to evaluate the impact of HTM structure on PSCs performance is the main objective of this manuscript. All the compounds (H1-H4) were successfully and characterized spectroscopically. Among these derivatives, author concluded that the derivative H3 exhibited better performance as HTM in the given cell structure of ITO/SnO2: PCBA/MAPbI3/HTM/MoOx/Ag. All the available data such as morphology, optical and thermal properties of H3 based cells supported the above said phenomena. Based on the novelty and achievement, this manuscript should be “ACCEPTED” to publish in your journal “International Journal of Molecular Sciences”. However, the following factors need to be clarified prior to the publication.
1. Author stated that “The detailed synthetic and purification procedures are described in Supporting Information”. However, there were no procedures or purification process found in the supporting information.
2. Why author selected alkyl chain lengths of H1-H4 as octyl and ethylhexyl?. Author said, it resembles the morphology of P3HT (hexyl substituted). Why author didn’t choose hexyl as alkyl chain lengths?
3. Suggestion: Every derivative has octyl and ethylhexyl groups at the terminal of thiophene. Instead, R1=R2=Octyl (H1), R1=R2=ethylhexyl (H2), then it is easy to identify the effect of varying alkyl chain lengths towards the performance.
4. For studying the efficiency, MoOx and Ag were used; whereas for stability and AFM, author used VOx and Al. However, the compatibility of the designed HTM (H1-H4) towards MoOx and VOx will not be same. How author can correlate these factors?
5. In DSC, why H2 and H4 didn’t show any crystallization peaks at cooling cycle, due to fluorine? Which means, the presence of fluorine affects the amorphous nature of those compounds?
6. The abbreviation for P3HT should be given in the manuscript.
7. In abstract second line, “main instrument” should be “main materials” or “one of the key materials”
8. For all the synthesized HTM, only spectral values were given, however no spectra were presented (even in supporting information too).
-After clarification of these factor, it can be accepted and published in your journal.
Author Response
The design and synthesis of structurally similar derivatives (H1-H4) based on alternating thiophene and benzothiadiazole (with/without fluorine) with varying alkyl chain lengths at the terminal thiophenes to evaluate the impact of HTM structure on PSCs performance is the main objective of this manuscript. All the compounds (H1-H4) were successfully and characterized spectroscopically. Among these derivatives, author concluded that the derivative H3 exhibited better performance as HTM in the given cell structure of ITO/SnO2: PCBA/MAPbI3/HTM/MoOx/Ag. All the available data such as morphology, optical and thermal properties of H3 based cells supported the above said phenomena. Based on the novelty and achievement, this manuscript should be “ACCEPTED” to publish in your journal “International Journal of Molecular Sciences”. However, the following factors need to be clarified prior to the publication.
Thank you for your positive evaluation of our work!
- Author stated that “The detailed synthetic and purification procedures are described in Supporting Information”. However, there were no procedures or purification process found in the supporting information.
Thank you, this statement was corrected.
- Why author selected alkyl chain lengths of H1-H4 as octyl and ethylhexyl?. Author said, it resembles the morphology of P3HT (hexyl substituted). Why author didn’t choose hexyl as alkyl chain lengths?
The observation about the morphology resembling that of P3HT was made after measuring AFM. Octyl and 2-ethylhexyl groups are widely used in designing of soluble organic semiconductor materials as more accessible side chains. We believe that the molecules with the same backbone and shorter hexyl substituents would be low soluble, thus their deposition using cheap solution -based techniques would hardly be possible.
- Suggestion: Every derivative has octyl and ethylhexyl groups at the terminal of thiophene. Instead, R1=R2=Octyl (H1), R1=R2=ethylhexyl (H2), then it is easy to identify the effect of varying alkyl chain lengths towards the performance.
In the presented set, all four materials comprise one of each substituents. For example, for H1 and H2 R1=octyl, R2=2-ethylhexyl, and for H3 and H4 the positions are vice versa: R1=ethylhexyl, R2 = octyl. The corresponding designation is used in Figure 1.
- For studying the efficiency, MoOx and Ag were used; whereas for stability and AFM, author used VOx and Al. However, the compatibility of the designed HTM (H1-H4) towards MoOx and VOx will not be same. How author can correlate these factors?
Thank you for this comment! According to our previous results (Energies 2021, 14, 5115, https://www.nanoge.org/proceedings/HOPE-PV/5f99ddbb6329f4157f4f2668), the nature of the metal oxide doesn’t influence the performance of organic HTM in devices. The main feature of VOx is that the three times thicker layer can be used (compared to MoOx) without losses in the device efficiency.
- In DSC, why H2 and H4 didn’t show any crystallization peaks at cooling cycle, due to fluorine? Which means, the presence of fluorine affects the amorphous nature of those compounds?
Actually, fluorine-containing compounds H2 and H4 showed both melting and crystallizations peaks on DSC curves (see Figure S1b). This suggest enhanced crystallinity of these small molecules as compared to non-fluorinated H1 and H3 counterparts, which showed signals just on heating curve.
- The abbreviation for P3HT should be given in the manuscript.
Thank you for noticing, the abbreviation was given
- In abstract second line, “main instrument” should be “main materials” or “one of the key materials”
Thank you for the suggestion, the corrections were made. We rephrased this statement as follow “The introduction of hole-transport materials (HTM) into the device structure is a key approach for enhancing efficiency and stability of devices.” (line 21)
- For all the synthesized HTM, only spectral values were given, however no spectra were presented (even in supporting information too).
Thank you for the comment! The NMR spectra were added as Figures S5-S20 in Supporting Information.
-After clarification of these factor, it can be accepted and published in your journal.
Many thanks for your comments and suggestion, which allowed to improve our manuscript!
Reviewer 4 Report
The article titled “Impact of backbone fluorination and side-chains position in thiophene-benzothiadiazole-based hole-transport materials on performance and stability of perovskite solar cells” presents four new small molecules and their application as HTMs in PSCs. The authors make a good characterization of the devices and set the objectives fairly clearly.
Next, I am going to make some recommendations and remarks:
1. English sholuld be revised and improved. For example, it is more natural to write "To avoid" than "For avoiding" (line 43).
2. The authors do not highlight the fact of not using dopants, which is usually very convenient for this type of device.
3. If the authors wish to use the denomination DADAD, they must establish what it means the first time they use it. (line75)
4. Synthetic procedures (line 240) and device fabrication (line 270) are not in the Supporting Information part.
5. This work would be more complete if H1-4 were compared with dopant-free spiro-OMeTAD.
6. I do not agree with the sentence “The more pronounced PL quenching of MAPbI3/H1 and MAPbI3/H3 was observed, which suggests potentially lower recombination rate on the interface between perovskite and non-fluorinated HTMs.”, in any case, the authors must include bibliography to support such affirmation.
Author Response
The article titled “Impact of backbone fluorination and side-chains position in thiophene-benzothiadiazole-based hole-transport materials on performance and stability of perovskite solar cells” presents four new small molecules and their application as HTMs in PSCs. The authors make a good characterization of the devices and set the objectives fairly clearly.
Thank you for your positive comments and suggestions!
Next, I am going to make some recommendations and remarks:
- English sholuld be revised and improved. For example, it is more natural to write "To avoid" than "For avoiding" (line 43).
Thank you, we checked the language and corrected the text
- The authors do not highlight the fact of not using dopants, which is usually very convenient for this type of device.
Thank you for pointing this out! We added the relevant information in Results and Discussion section (lines 305-312).
Actually, such dopants as bis(trifluoromethane)sulfonimide lithium salt (LiTFSI), 4-tert-butylpyridine (t-BP) and other are widely used for improving the conductivity of charge transport materials in perovskite solar cells and, hence the performance of devices [Sci. Rep. 2012 591; ACS Energy Lett. 2018, 3, 7, 1677–1682]. However, these reagents accelerate degradation of devices [Adv. Mater., 2016, 28, 10738-10743; Nano Energy 2021,82, 105751]. Therefore, we did not used dopant reagents in order to provide long-term stability of photovoltaic cells.
The revised text is the following:
Low molecular weight organic HTMs are usually applied with dopants for enhancing their charge transport characteristics at the price of device stability mitigation. [Sci. Rep. 2012 591; Adv. Mater., 2016, 28, 10738-10743; Nano Energy 2021,82, 105751] It is worth noting, that the presented results are achieved for undoped HTMs.
- If the authors wish to use the denomination DADAD, they must establish what it means the first time they use it. (line75)
Thank you for this suggestion! We added the explanation for DADAD abbreviation.
Herein, we present four novel conjugated small molecules with DADAD structure (D-donor thiophene block, A – acceptor benzothiadiazole block) and their investigation as HTMs in perovskite solar cells (Figure 1).
- Synthetic procedures (line 240) and device fabrication (line 270) are not in the Supporting Information part.
Thank you for pointing this out! We corrected this sentence! The details of synthesis and device fabrication are provided in Materials and methods section.
- This work would be more complete if H1-4 were compared with dopant-free spiro-OMeTAD.
Thank you for this suggestion! We provided the performance of reference PSCs based on non-doped spiro-OMeTAD (Figure S5, Supporting Information) and relevant description in main text (lines 303-310). The PCEs of these cells were 7.0±2.0%, with VOC=910±60 mV, JSC=16.1±0.5 mA cm-2, FF=46±10 %. These results are in agreement with previously reported findings [J. Power Sources 2017, 342 (28) 886-895; J. Energy Chem. 2021,55, 211–218] and could be attributed to low hole mobilities/conductivity of dopant-free spiro-OMeTAD [Appl. Mater. Interfaces 2018, 10, 14, 11633–11641]. It should be also noted that the synthesis of spiro-OMeTAD is more complex than of presented oligomers, which makes them promising HTMs for perovskite solar cells.
- I do not agree with the sentence “The more pronounced PL quenching of MAPbI3/H1 and MAPbI3/H3 was observed, which suggests potentially lower recombination rate on the interface between perovskite and non-fluorinated HTMs.”, in any case, the authors must include bibliography to support such affirmation.
We agree with the Reviewer that PL quenching alone does not allow to judge in full about recombination of charge carriers. It rather shows the ability of HTM layer to extract the holes from perovskite absorber material [Organic Electronics, 2019, 74, 7-12; J. Energy Chem. 2021,55, 211–218]. We revised aforementioned conclusion in the following manner: The more pronounced PL quenching of MAPbI3/H1 and MAPbI3/H3 was observed, which suggests better hole extraction from perovskite to non-fluorinated HTMs. (lines 313-315)
Round 2
Reviewer 1 Report
The authors addressed most of the comments satisfactorily. However, high-performance HTLs reported in literatures were not discussed and cited although they mentioned that the new references are cited. Perhaps, authors might forget to do so while they pay more attention to other comments.
Author Response
The authors addressed most of the comments satisfactorily. However, high-performance HTLs reported in literatures were not discussed and cited although they mentioned that the new references are cited. Perhaps, authors might forget to do so while they pay more attention to other comments.
We apologize for the unpleasant situation! This was indeed overlooked. The references are cited now (references 18-21).